# Preparation and Characterization of Corn Starch Bio-Active Edible Packaging Films Based on Zein Incorporated with Orange-Peel Oil

**DOI:** 10.3390/antiox8090391

**Published:** 2019-09-11

**Authors:** Yue Wang, Rong Zhang, Saeed Ahmed, Wen Qin, Yaowen Liu

**Affiliations:** 1College of Food Science, Sichuan Agricultural University, Yaan 625014, China; yueerjiejie520@163.com (Y.W.); zhangronglzy@163.com (R.Z.); saeedahmed_mahar@yahoo.com (S.A.); qinwen@sicau.edu.cn (W.Q.); 2School of Materials Science and Engineering, Southwest Jiaotong University, Chengdu 610031, China; 3California Nano Systems Institute, University of California, Los Angeles, CA 90095, USA

**Keywords:** corn starch, orange-peel oil, zein

## Abstract

Zein, corn starch (CS), and orange-peel oil (OPO) extracted from orange peels were used to prepare novel corn starch/orange-peel oil/zein nanocapsules (OZN) bio-active food packaging materials. The results showed that the OZN were round, smooth and in compact morphology with an average diameter of 102.7 ± 10.5 nm from OPO and zein (3:10, *w/w*). By testing the turbidity and atomic force microscopy (AFM) of OZN and the mechanical properties and water vapor permeability of the composite films, the comprehensive properties of composite films with different mass ratios were analyzed. It showed that the addition of OZN improved the mechanical and moisture barrier properties and extended the release time of OPO. When the ratio of OZN and CS is 5:5, the highest elongation at break and tensile strengths is achieved, at values of 30.91% ± 2.52% and 12.19 ± 1.97 MPa respectively. The relative release concentration of OPO was highest at a ratio of 5/5, and over time it would last longer to maintain a higher release concentration. Besides, the oxidation resistance of the composite film was good, especially when the ration of starch CS to OZN was 5/5, it had the highest DPPH radical scavenging activity (30.16% ± 1.69%). Thus, it can be used as a bio-active edible food packaging film to ensure the safety of food products and reduce environmental pressure to some extent.

## 1. Introduction

In recent years, the full use of plastic food packaging materials has brought about increasingly severe challenges to the global ecological environment. Meanwhile, growing awareness of food safety has made biodegradable materials more and more popular [1]. Due to being less-polluting to the environment and their abundance sources in nature [2,3], many natural macro-molecular materials are being applied in food packaging, such as starch, chitosan, cellulose, silk fibroin protein, gelatin, among others [4,5]. Starch is one of the natural macro-molecular materials with abundant natural reserve and biodegradability. Due to its easy availability, low-cost and functional characteristics, it is often used as a thickener, a gelling agent, and stabilizer [6,7]. Among the materials, corn starch (CS) is increasingly applied in food, medicine, paper-making, etc. [8], due to its richness, low cost, stable yield, biodegradability, good hydrophilicity and many other advantages, especially in the research of edible starch film [9]. But its application is limited due to the insufficient mechanical properties and barrier properties of pure CS film [10]. Slavutsky et al. mixed it with levan for modification, and the composite film properties were improved. Therefore, to meet the requirements of application, it is necessary to change or add some other materials in the current study of starch-based composite materials [11,12].

CS was added into the matrix of polycaprolactone (PCL)/pomegranate peel extracts to prepare antibacterial packaging materials and it was found through bacteriostatic experiments that PCL/CS/pomegranate (PR) composite film had an excellent inhibitory effect on Staphylococcus aureus [13]. CS not only reduced the cost, but also increased the rigidity of the PCL matrix, and it provided a release channel to release polyphenols by weakening the interaction between PCL and PR, and enhancing the antimicrobial activity of PR. However, traditionally blending active materials into the composite film could not control the release of active compounds from the films. Due to the volatility of the active material, an active solid film was prepared by plasma polymerization [14], and the active material can also be encapsulated to achieve a sustained release effect. A preparation method of the oil–core zein–shell micro-capsules by the precipitation of zein from the continuous phase of the oil-in-(water/ethanol) emulsion onto the oil droplets without any reactant has been reported [15]. An innovative study which aimed to develop zein-based core-shell microencapsules loaded with limonene to protect it and control its degradation was successful [16]. While there has been no report about the form of starch-based bio-active edible packaging films based on zein with orange-peel oil (OPO) in food packaging, natural antimicrobial compounds have been used to ensure food safety and extend expiration dates. Orange peels are highly biodegradable and have high value-added, which contain many valuable compounds including carbohydrate polymers, fermentable sugars, polyphenols flavonoids, and essential oils [17]. OPO has many health benefits and is highly resistant to oxidation. The main component of OPO is limonene [18], which has many advantages; for example, it is a colorless oily liquid with a fresh citrus flavor and broad-spectrum antibacterial activity [19]. However, many valuable compounds in OPO, especially limonene, can suffer oxidative degradation under normal storage conditions. Thus, encapsulating OPO with emulsions systems is an effective method to improve its physical stability. Compared with other colloidal systems (e.g., liposome, micelles, and so on), zein has more advantages. Zein is the primary storage protein in corn and accounts for nearly half of the total protein of the corn grain, more than 50% of zein is hydrophobic amino acid residues, which helps to delay the release and oxidation of biologically active compounds. It has been reported that zein can be assembled into varieties of structures easily because of its amphiphilic nature. And the conjugates or complexes formed by the combination of zein and other compounds are also easy to fabricate which allows zein to work as a delivery system to transport compounds.

In this experiment, OPO from orange peels was extracted and made into various concentrations under different process conditions. Then, the corn starch/orange-peel oil/zein nanocapsules (OZN) were prepared in order to find optimal mixing proportions. Finally, the composite film was prepared by the casting film formation method. The thickness, density, mechanical properties, relative release antibacterial activities of the composite films were compared, and optimum conditions were established and its application was analyzed.

## 2. Materials and Methods

### 2.1. Materials

The glycerin (purity 99.97%, density 1.26 g/cm^3^) used in this study was supplied by Chengdu Kelong Chemical Reagent Factory. The Anhydrous ethanol (effective substance content up to 99%) was also supplied by it. The CS was ordered from Chengdong Food Co., LTD in Shuangliu County of Chengdu. Orange used in this study were from a spontaneous mutation of the cultivar produced in Fengjie County (Sichuan). All above was of edible grade, and all chemicals and solvents were of reagent grade or higher purity, which were purchased from Yan’an Wanke Reagent Co., China, unless otherwise indicated.

### 2.2. Ultrasound-Assisted Extraction of Orange-Peel Oil

Oranges were collected and decorticated to gather the peels. Then a number of peels and 200 g of ethyl acetate were added into water and stirred while heating. When the temperature rose to 30 °C, the ultrasonic device was turned on and kept working for 10–30 min. After the ultrasonic removal, 2 g of activated carbon was added into the mixture and stirred it for about 30 min. When the solution was cooled down to 15 °C, the mother liquor was filtered under reduced pressure and the solid waste was obtained by filtration. The filtrate was layered into an organic phase and an aqueous phase. The aqueous phase was extracted again with 20 g of ethyl acetate, and the organic phase to give additional. The combined organic underwent two steps, the organic phase of atmospheric distillation and ethyl acetate phase to obtain a pale yellow to colorless oil residue. The OPO was collected, dried over anhydrous sodium sulfate and stored at 4 °C until being used. Yield of OPO was evaluated as follows:(1)$$\mathrm{OPO}\;\mathrm{yield}\;(\%)=\frac{\mathrm{weight}\;\mathrm{of}\;\mathrm{OPO}\;\mathrm{obtained}\;\mathrm{after}\;\mathrm{extraction}}{\mathrm{weight}\;\mathrm{of}\;\mathrm{orange}\;\mathrm{peels}\left(\mathrm{dry}\;\mathrm{matter}\right)}\times 100$$

In order to optimize the extraction process conditions, OPO extraction experiment was designed according to Table 1, and were performed at different ultrasonic times (30, 40, and 50 min), ultrasonic power (80 W, 90 W, and 100 W) and the ratio of orange skin/ethyl acetate solution (1:6, 1:8 and 1:10). To this purpose, the optimal extraction conditions were explored with viscosity and yield of OPO being indicators.

### 2.3. Preparation of OZN

OPO and zein were weighed in different radios of 1/10, 2/10, 3/10, 4/10, 5/10, and zein was dissolved in 50 mL of 80% aqueous ethanol and stirred at 350 rpm for 1 h at room temperature. 0.1% (*w/w*) Tween 20 at a ratio of 10% of the total weight was incorporated to the solution under a magnetic stirrer. OPO was added into this solution drop-wise gradually. Mixed emulsions were finished using an ultrasonic homogenizer (model T-25, IKA Instruments, Germany) at applied power of 400 W, 20 kHz, 12 mm probe diameter) at 25 °C (temperature was controlled by a container of ice and water) and the power of 350 W for 10 min. OZN for testing was obtained by steps of rotary evaporation at 60 °C for 15 min and freeze-drying.

### 2.4. Turbidity Test

Before the test, 1 g of OZN powder was dissolved in 80% ethanol at room temperature for 24 h. The turbidity was determined by using a UV–visible spectrophotometer at 600 nm (Epoch, BioTek Instruments, VT, USA) by the transmission of light through a 2-mm path-length cuvette.

### 2.5. The Encapsulation Efficiency of OPO

The content of OPO in OZN was determined and calculated as follows: 1 g of OPO–zein nanocapsules was dissolved in 20 mL N, N-dimethylformamide (DMF) and 20 mL deionized water was added, then mixed for 1 min. OPO was extracted with DMF by heating the sample in glass tubes at 45 °C in a water bath with intermittent mixing. The cells were cooled to room temperature and DMF was separated from the aqueous phase by centrifugation at 4000 rpm for 20 min [20]. The amount of OPO present in DMF was quantified by measuring absorbance at 252 nm by a UV–visible spectrophotometer (Epoch, BioTek Instruments, VT, USA). The calculation formula is as follows:(2)$$Q=\frac{N_{1}-N_{2}}{N_{1}}\times 100\%$$
where N_1_ is the mass of the OPO before forming nanocapsules; N_2_ is the mass of OPO in OZN.

### 2.6. Preparation of CS/OZN Composite Films

To prepare CS/OZN solutions, first, the CS was placed in a water bath at 80 °C for 1 h to complete gelatinization. After the film liquid was cooled down to room temperature, the OZN were mixed with CS solutions, which were weighed at mass ratios of 10/0, 1/9, 3/7, 5/5, and 7/3. Then 1 wt% glycerin was added to the solutions as a plasticizer [21,22]. At the last, 80 mL of the measuring cylinder was uniformly poured onto the glass, and placed in an oven at 35 °C drying for 6–8 h. Figure 1 shows a schematic diagram of the steps for preparing CS/OZN films used in the experiments.

### 2.7. Film Thickness 

The thickness of each composite film was measured by a thickness gauge, and 10 points were uniformly taken at the center and the periphery of the sample to the nearest 0.01 mm, and the average value was taken as the thickness value *t* of the composite film. The thickness of the composite film is determined to provide a basis for the determination of other indicators [23].

### 2.8. Light Transmission and Color Properties 

The light transmission test performed in the visible light region and film color properties were evaluated by using the CIELAB color space by means a KONICA CM-3600d COLORFLEX-DIFF2, HunterLab, Hunter Associates Laboratory, Inc, (Reston, Virginia, USA). Color coordinates, L (lightness), a* (red–green) and b* (yellow–blue) were measured. The instrument was calibrated with a standard white tile. Measurements were carried out in quintuplicate at random positions over the film surface. Average values for these five tests were calculated. Total color difference (ΔE) was calculated as follows:(3)$$\Delta E=\sqrt{{\Delta a}^{2}+{\Delta b}^{2}+{\Delta L}^{2}}$$

### 2.9. Oxygen and Water vapor Transmittance Test

Nitrogen was used as the oxygen carrier gas (relative humidity ∼50%) at a temperature of 23 °C; the oxygen and nitrogen flow rates were 20 and 10 mL/min, respectively. The water vapor transmission coefficient of the composite film was measured by a PERMETM W3/031 water vapor transmission rate tester and oxygen transmission rate test was measured by a PERME TM OX2/231 permeability tester [24].

### 2.10. Mechanical Performance Test

Tensile strength and elongation at break of composite films were tested according to ASTM D828-97 standard test method [25]. Before testing, the composite film was cut into strips of 10 × 150 mm with an initial clamping distance of 50 mm and a test speed of 50 mm/min. Tensile strength and elongation at break were calculated as follows:TS = LP/A × 10^−6^(4)
where TS is the tensile strength, MPa; LP refers to the maximum tension sustained by the film fracture, N; A is the active area of the film, m^2^.
E = ΔL/L × 100%(5)
where E is the elongation at break of the film, %; ΔL is for film rupture length added value, mm; L is the original sufficient length of the film, mm.

### 2.11. SEM Analysis 

To obtain more details, the surface morphology of the composite film was analyzed by SEM (FEI Quanta 200, The Netherlands), and a 3 × 3 mm square was cut from the sample composite film for scanning.

### 2.12. AFM Analysis 

AFM (DualScopeTM DS95-50, DME, Denmark) was used to verify the morphological characterization and particle size distribution of OZN. AFM images were recorded in tapping mode (AFM Arrow Cantilever, Reflex-coated; 2.8 N/m, 75 kHz) using a WITec alpha 300 RSA + microscope (Ulm, Germany). To prepare the optimum sample for AFM imaging, 1 µL droplets of unloaded OZN in ultrapure water were put on CaF_2_ substrate and dried at room temperature for 30 min. Then the image analysis by AFM was performed [26]. 

### 2.13. FTIR Analysis

A FTIR (NICOLET Is10, Suzhou Junquan Instrument Co., Ltd.) instrument was used to collect the infrared spectra of the composite film surface. The scanning frequency was set at 32 times/s, the resolution was 4 cm^−1^, and the environment temperature was 25 °C, to measure the precision of wavenumber in 4000–500 cm^−1^ of the absorption spectrum. The collected spectra were analyzed by OMNIC8.0 software.

### 2.14. Relative Release of OPO from CS/OZN Composite Films

OPO released content was quantified by measuring absorbance at 252 nm by a UV–visible spectrophotometer (Epoch, BioTek Instruments, VT, USA). Aqueous solutions containing 95% of ethanol were used as the liquid food simulant, and films of 8 × 8 cm were introduced into 11 mL vials, containing 2 mL of each food simulant, which were hermetically closed and slightly stirred during storage. Samples (100 μL of simulant solution) were taken with a syringe through a septum in each vial for testing. The relative release of OPO was calculated as follows:(6)$$Q=\frac{N_{2}-N_{3}}{N_{2}}\times 100\%$$
where N_2_ is the mass of OPO in OZN; N_3_ is the release of CS/OZN composite films.

### 2.15. DPPH Radical Scavenging Assay

A DPPH free radical scavenging assay was carried out according to previous research with slight modification [27]. Twenty-five mg of film sample was immersed in 4 mL ethanol to get the extract. Consequently, 4 mL of film extract was added to 1 mL of 0.1 mm ethanolic DPPH solution. It was then mixed thoroughly and was incubated in a dark room at normal room temperature for 30 min. The absorbance was measured at 517 nm. The DPPH radical scavenging activity was measured using the following equation:(7)$$\mathrm{Radical}\;\mathrm{scavenging}\;\mathrm{activity}\;\left(\%\right)=\frac{\left(\mathrm{Abs}\;\mathrm{DPPH}-\mathrm{Abs}\;\mathrm{sample}\;\mathrm{extract}\right)\times 100}{\mathrm{Abs}\;\mathrm{DPPH}}$$

### 2.16. Statistical Analysis

All preparations were carried out in triplicate, statistical differences (*p* > 0.05) were calculated using analysis of variance—Tukey’s.

## 3. Results

### 3.1. Effect of Different Ultrasonic Processes on the Extraction of OPO 

To optimize the extraction process of OPO, an orthogonal experiment was designed to extract OPO at room temperature. The extraction rate of it was used as the main index, and the viscosity of the obtained product was tested at the same time, and each sample was tested three times. The results are shown in Table 2. From the perspective of ultrasonic power, when the ultrasonic power was lower than 90 W, the extraction rate of OPO increased with the growth of ultrasonic power with its highest at 90 W. This may be caused by the fact that the solvent entered the solid matter quickly and dissolved the OPO contained in the solution as entirely as possible, and strengthened the extraction and separation process, thereby effectively increasing the yield of OPO [28]. When the power exceeded 90 W, the extraction rate of OPO decreased slightly with the increase of ultrasonic power. It may be brought about by the fact that when the power was increased, the solvent temperature was significantly increased, and the evaporation of OPO was increased. From the duration of ultrasound, when the ultrasonic time was lower than 40 min, the extraction rate of OPO increased with the growth of ultrasonic time, and the extraction rate was the highest at 40 min. This may be because an increase in the ultrasonic time enabled the OPO to be sufficiently dissolved within a specific range. When the time exceeded 40 min, the extraction rate of OPO decreased slightly with time, which was caused by the high temperature of the solvent [29]. Therefore, when the amount of evaporation of OPO increased, its yield would drop. The analysis found that the effects of ultrasonic power and ultrasonic time on the extraction rate of OPO were similar [30]. It would cause temperature to change during the ultrasonic process and the temperature directly affected the cavitation of the ultrasound [31,32]. The proper increase of the ultrasonic temperature was beneficial to the cavitation, which helped to increase the extraction rate of essential oils. However, when the ultrasonic temperature was too high, the vapor pressure in the bubbles increased, and when the bubbles were closed, the buffering effect was promoted, and the cavitation was weakened, thereby reducing the extraction rate.

From the ratio of material to liquid, when the ratio of plastic to liquid was 1:8, the extraction rate reached a higher value, and as the solvent increased, the extraction rate did not increase significantly. This may be because the solvent extraction was the limiting factor which had been removed when the feed to liquid ratio was 1:8 [33]. If the content of the material was too high, the diffusion process of OPO in water would be subject to higher mass transfer resistance. If the liquid ratio was too high, the amount of OPO dissolved in water would increase, which would also affect the extraction rate. Therefore, the optimal extraction process of OPO was: Ultrasonic power at 90 W, ultrasonic time at 40 min and the material–liquid ratio at 1:8.

### 3.2. Encapsulation Rate and AFM and SEM of OZN

OPO/zein in the preparation of nanocapsules was at the mass ratios of 1/10, 2/10, 3/10, 4/10, and 5/10 respectively, and the encapsulation results are shown in Figure 2. It can seen from the figure that the encapsulation effect was different, probably because of the different mass ratios of OPO and zein, the thickness of the nanocapsule core–shell formed during the precipitation process was different. The best encapsulation effect occurred at a ratio of 3/10, with encapsulation rate reaching at 71.74% ± 1.13%. At the ratios of 4/10 and 5/10, the encapsulation effect was also not bad. The encapsulation effect increased first and then decreased as the proportion increased, suggesting that the mass ratio of OPO and zein could increase the encapsulation effect of nanocapsules within a certain range. When the content of OPO was too much, the encapsulation efficiency decreased, indicating that zein was insufficient to encapsulate all OPO, thus resulting in an aggregation of OPO droplets and formation of solid zein spheres. When there is too much zein, it could be inferred that zein was sufficient to form a complete shell around the OPO droplets, which in turn grew thicker due to further deposition of zein from the bulk. The excess zein would be self-assembled and precipitated in the form of solid particles due to the decrease of zein solubility with water addition, confirmed by Filippidi et al. [34] who reported that addition of water to the initial zein solution resulted in homogeneous precipitation and the formation of solid particles of pure zein. Therefore, the OPO/zein ratio of 3/10 showed the best encapsulation effect. The spherical appearance and nanostructure of the freeze-dried OPO–zein-3/10 nanocapsules were observed by AFM (Figure 3a) and SEM (Figure 3b). It was revealed that most of OPO–zein-3/10 nanocapsules had a particle size of 102.7 ± 10.5 nm. There were also very few relatively small particles, possibly self-assembling particles formed by zein. Also, the composite nanocapsules had a uniform particle size [35], no agglomeration, and exhibited good dispersibility, which indicated that this system was capable of producing nanocapsules with almost the same mean particle size and resolved the common problems of previous studies, such as flocculation.

### 3.3. Turbidity Test

The turbidity of an emulsion is a function of droplet size and concentration. Therefore, the change in turbidity indicates the change in droplet size and concentration with time. Figure 4 shows the turbidity of different ratios of OZN after 24 h. It could be seen from the figure that as the ratios increased, the turbidity of the emulsion of OZN first decreased and then increased, which mainly changed in the range of 0.37 ± 0.11 to 0.72 ± 0.15. When the ratio was 3/10, the turbidity was the lowest at 0.37 ± 0.11. When the ratio was further increased, although the turbidity increased, the increase was not obvious, indicating that the balance of Van der Waals, electrostatic, and polymeric steric interactions were the main droplet interactions playing a role in determining the stability of emulsion systems. The magnitude of repulsive interactions and forces between emulsion droplets plays a positive role in nanoemulsion stability. However, flocculation, coalescence, and aggregation may be responsible for the increased turbidity when the ratio was further increased [36].

### 3.4. Light Transmission and Color Properties

Table 3 shows that when the ratio of CS to OZN was 10/0, the highest transmittance was 93.07% ± 1.32%, as the ratios increased, more OAN formed, which made the structure more compact. Light transmittance, in general, was on the decline compared with pure CS film [37], but the drop degree was not big. When the ratio of CS to OZN was 5/5, the light transmittance was 87.29% ± 1.12%, which only reduced by 6.21% compared with pure CS film. Additionally, the light transmittance decreased, which may be the reason that the dense structure was destroyed due to the excessive density of the OZN [38,39].

Regarding the color of the film, Figure 5a shows the visual appearance of the composite film. The pure CS film was used as a control group, and the mass ratio of 10/0, 9/1, 7/3, 5/5, and 3/7 of composite films was measured. As expected, the pure CS film was colorless and transparent, and after adding OZN, a pale yellow film was obtained. A slight decrease in the luminance value (L*) was observed. However, both pure CS film and composite film samples exhibited high L* values, indicating their high brightness [40]. All in all, compared with pure CS film, despite the difference in color and light transmittance, it maintained high transparency and could meet consumer requirements for food packaging materials [41].

### 3.5. Properties of Varying CS/OZN Composite Films

The mechanical properties and barrier properties of the composite film were better than those of the pure CS film, it shows that the composite film has good resistance to elongation and load bearing capacity, as can be seen from Table 4. Due to the increase of OZN, the tensile strength of the composite film changed little but the elongation at break was significantly improved [42,43]. It showed that OZN had the effect of plasticizer, which made the composite film have lower brittleness than pure CS film; OZN can be inserted into the matrix to replace the interaction between some molecules in the composite film and form a hydrogen bond with the hydroxyl group of CS, thus changing the mechanical properties of the composite film, which agreed with Arrieta et al. [41]. It is well known that films used for packaging require a high degree of flexibility to avoid cracking during processing and use [44]. 

It was found that the elongation at break of the composite film was generally higher, the highest elongation at break reached 42.91% ± 3.16%. This was due to the plasticization effect achieved by the addition of OZN, and the results showed that the flexibility of the CS film was significantly improved, which was consistent with the conclusions drawn by Cano et al. [45]. The addition of OZN made the molecule more stable, with less steric hindrance and better compatibility in the substrates, thus significantly increasing the flexibility of the composite film. When OZN content was further increased, the molecular chain spacing may have increased, thus reducing elongation at break [46]. With the increase of OZN, the tensile strength improvement effect was significant because the minimum tensile strength increased from 9.42 ± 1.06 to 12.19 ± 1.98 MPa. When the CS to OZN was 7/3, 5/5, 3/7, the oxygen transmission rate all appeared at a low level. With the increase of OZN, the water vapor transmission rate of the composite film was gradually lowered, and the moisture resistance was increased. One reason is that OZN retained the excellent properties of OPO and had unique hydrophobic properties, as found by Ribeiro Santos et al. that essential oils can reduce water vapor transmittance (WVP) [47]. Its hydrophobicity limited the formation of hydrogen bonds between the CS matrices. Zein is moderately hydrophobic due to its high content of non-polar amino acids which introduce a better barrier to moisture and an excellent barrier to oxygen [48]. In general, the moisture barrier properties of the pure CS film need to be further improved. The essential oil-based hydrophobic material can be added to modify it to improve the moisture barrier properties. The other reason is that OZN retained the excellent properties of zein and had unique hydrophobic properties [49]. On the other hand, the increase of OZN enhances the interaction between the various films of the composite film, increases the intermolecular force, and makes the composite film structure denser, which agreed with the result of Dai et al. [50].

### 3.6. SEM Analysis

SEM analysis was performed on the composite films with pure CS film (a), and CS/OZN composite films which were at the mass ratios of 10/0, 9/1, 7/3, 5/5, and 3/7 [51]. It can be preliminarily seen from Figure 5 that OZN is uniformly dispersed in the matrix and has good compatibility with the composite film. The content had a positive influence on the morphology of the composite film. Within a certain range, the composite film structure was relatively dense, without pores, bubbles, or gaps. When the OZN content continued to increase, excessive OZN damaged the compact structure of the composite film and may accumulate on the surface of the composite film, resulting in different degrees of cracks on the surface [52]. Also, the sample preparation process was also a reliable guarantee to obtain clear, ideal, and real SEM observations.

### 3.7. FTIR Analysis

Infrared spectra of OPO, zein, OZN, CS/OZN composite films were obtained, as shown in Figure 6. The spectrum of OPO showed a characteristic peak at 886 cm^−1^ due to C–H bending, 1645 cm^−1^ due to C=C stretching, and 2826 and 2971 cm^−1^ due to C–H stretching [53]. The band in 3100–3500 cm^−1^ was due to the O–H stretching vibration of the hydroxyl group. In the spectra of zein and CS/OZN composite films, the characteristic peaks of the hydroxyl group were at 3306 and 3316 cm^−1^, respectively, which was similar to the result of Li et al. [54]. The first characteristic peaks of zein were 1658 cm^−1^ (amide I, C=O stretching) and 1538 cm^−1^ (amide II, N−H bending). Besides, in the spectra of OZN, both the characteristic peaks of OPO and zein appeared. The characteristic peak of the C–O–C bond is 1014 cm^−1^, and the absorption peak is blue-shifted to 1019 cm^−1^. This suggests that the addition of OZN may hinder the formation of hydrogen bonds in the matrix. Also, comparing to OPO, a new peak was found in the spectrum of CS/OZN composite films at 1655 cm^−1^, corresponding to the characteristic peak of zein. Similar results were also observed in the spectrum of zein/rhamnolipid composite nanoparticles [50].

### 3.8. Relative Release of OPO from CS/OZN Composite Films

In this study, the amount of OPO released relative to the composite film within 300 h was recorded. The results in Figure 7 show that the relative release concentration of OPO was the highest at a ratio of 5/5, and over time it would last longer to maintain a higher release concentration without sudden release after 120 h. This phenomenon was the same as the explosive release of essential oils which was recorded by some scholars [55]. This indicated that the composite film formed by nanoencapsulation of OPO in the food industry had controllability for the release of OPO, and the surfactants Span 80 may increase the fluidity between the components and regulate the permeation rate of OPO [56]. When the content of OZN was excessive over 5:5, the relative release concentration was lowered. The reason may be that excessive nanocapsules may agglomerate during film preparation, making it impossible to mix the nanocapsules with CS completely [57]. Some nanocapsules aggregated and the formed agglomerated nanocapsules were on the surface of the composite film (Figure 5), thereby volatilizing due to being at a surface and having a high temperature during baking, so that the final obtained composite film itself had a low content of OPO, and therefore, the relative release concentration was also low [16].

### 3.9. DPPH radical Scavenging Assay

Compared with the control, the films with OZN incorporated had significantly higher DPPH radical scavenging activity with increasing OZN concentrations than the control. The higher antioxidant activities of films mixed with OZN were because of the fact that OPO is a monoterpene compound (Figure 8). These results indicated that the antioxidant activity of films incorporated with OZN was significantly improved, and increased with the growth of the concentration of OZN. Especially when the ration of CS to OZN was 5/5, it had the highest DPPH radical scavenging activity (30.16% ± 1.69%), since orange-peel oil contains flavonoids. The CS/OZN composite film could improve shelf life and maintain the quality of foods, moreover, OZN may have potential health benefits for the consumers when the edible films are consumed and the OPO is released. In general, the addition of OZN into CS matrix films leads to an increase in the antioxidant activity of films. The extraction of terpenoids from the orange residue was performed in portable extractor equipment, the DPPH free radical scavenging rate was 12.1%, which also showed that it had certain antioxidant activity [58], and the results were similar to our experimental results. However, after fermenting the orange peels, the antioxidant activity of the fermentation medium extract reached 20.17%, indicating that the fermentation process improved the oxidation resistance of the terpenoids in the orange peels [59].

## 4. Conclusions

The main purpose of the study was to develop a modified corn film prepared by the casting film method after the OZN were finished by exploring the effects of the mass ratios of CS/OZN on the density, transmittance, mechanical properties, water vapor transmission rate, and oxygen permeability of composite films, combined with SEM analysis of composite films. The apparent performance of the composite film was found to be good. The light transmittance of composite films was decreased, the mechanical properties and barrier performance were improved greatly. The addition of OZN also extended the release time of OPO. As a result, a new idea was provided to control the sustained release of volatile oil, that is, it should be encapsulated first and then combined with the film-forming substance. Therefore, this experimental composite film is expected to be used as a new type of food packaging material, in particular for dry matter in the food industry to reduce the use of internal plastic packaging to alleviate ecological pressure and it has a broad prospect for development.

## Figures and Tables

**Figure 1 antioxidants-08-00391-f001:**
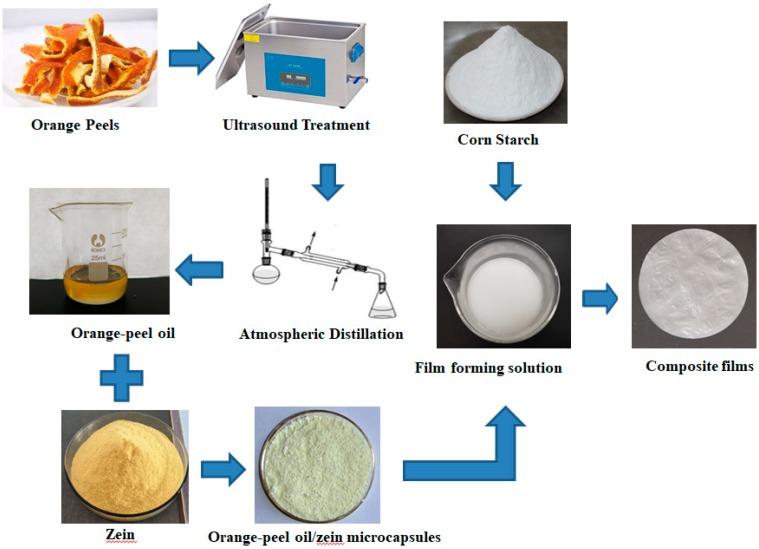
The schematic diagram of the steps to fabricate the corn starch (CS)/orange-peel oil (OPO)/zein nanocapsules (OZN) composite films.

**Figure 2 antioxidants-08-00391-f002:**
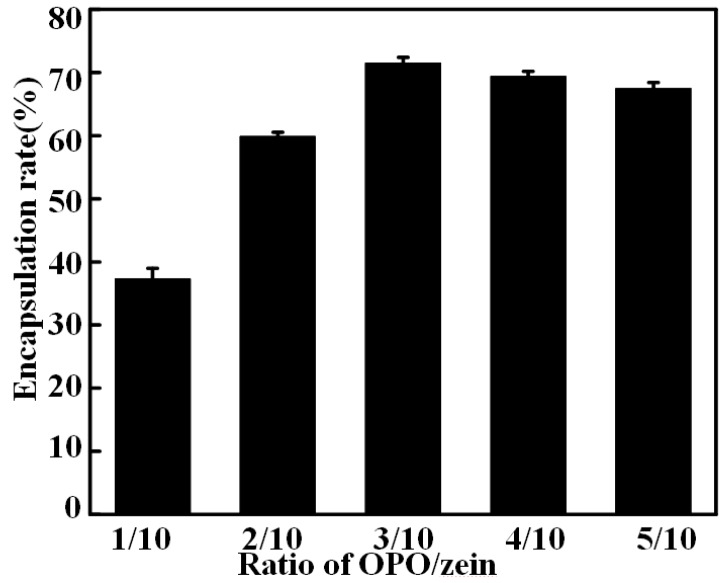
Encapsulation rate of different mass ratios of OPO/zein (1/10, 2/10, 3/10, 4/10, 5/10).

**Figure 3 antioxidants-08-00391-f003:**
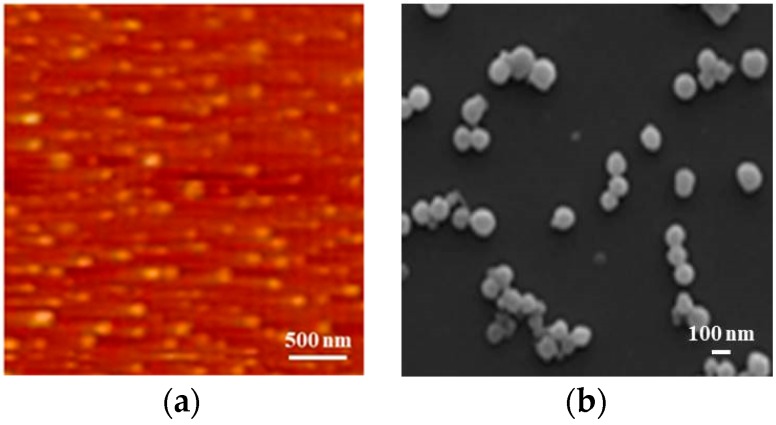
(**a**) AFM and (**b**) SEM images of the freeze-dried OPO/zein-3/10 nanocapsules.

**Figure 4 antioxidants-08-00391-f004:**
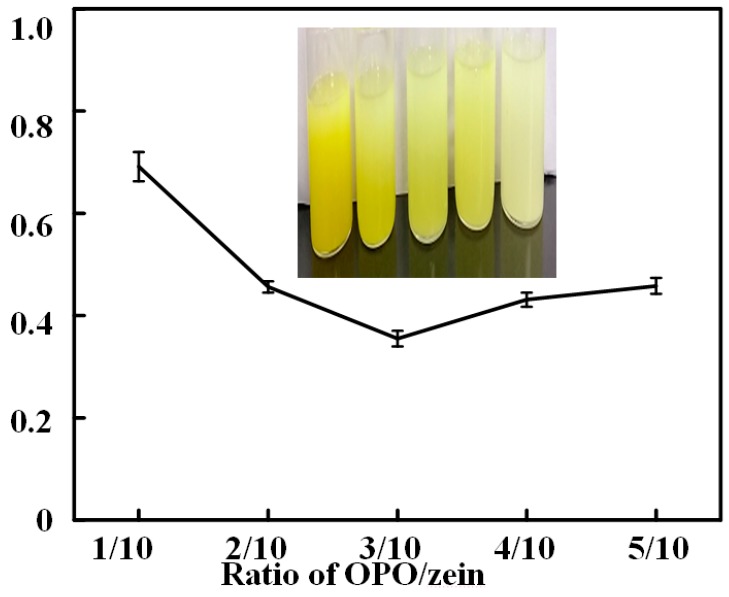
Turbidity of different OZN under 24 h at the room temperature.

**Figure 5 antioxidants-08-00391-f005:**
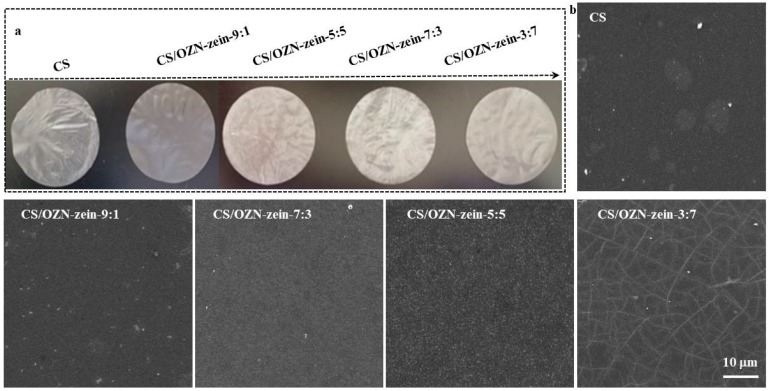
(**a**) Visual appearance of different CS/OZN composite films; (**b**) SEM images of different CS/OZN composite films.

**Figure 6 antioxidants-08-00391-f006:**
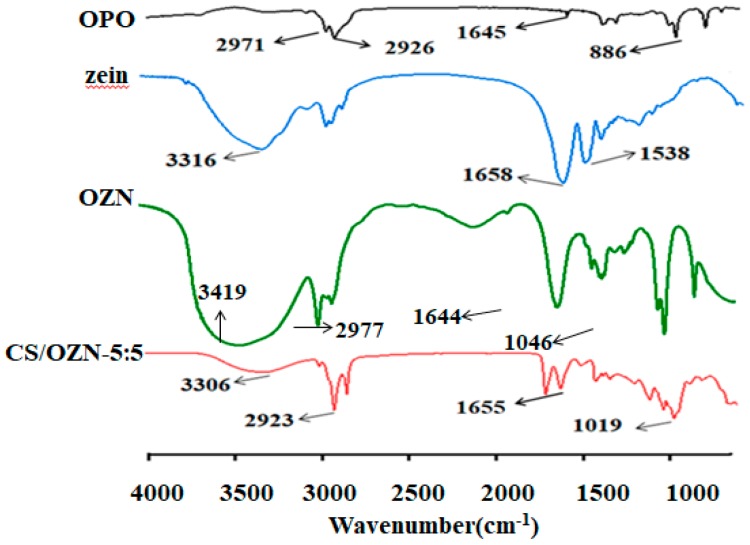
FTIR spectra of OPO, zein, OZN and CS/OZN-5:5 composite films.

**Figure 7 antioxidants-08-00391-f007:**
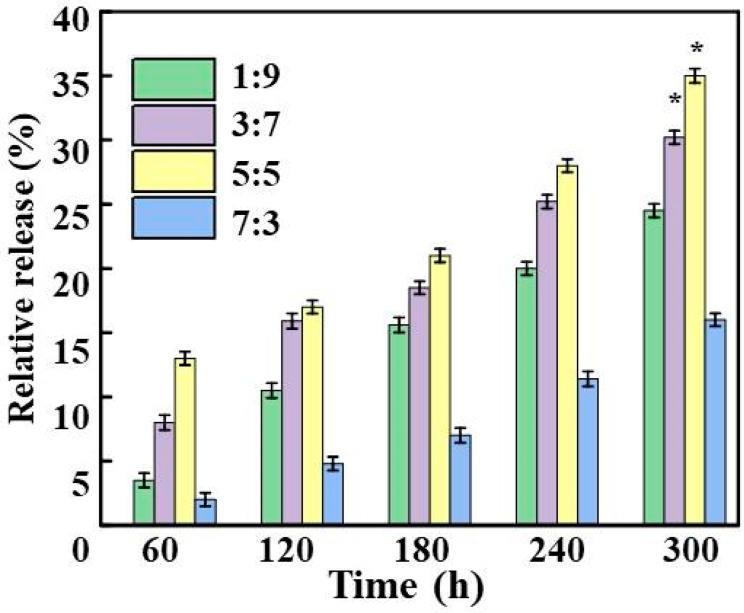
Relative release of OPO from different CS/OZN composite films.

**Figure 8 antioxidants-08-00391-f008:**
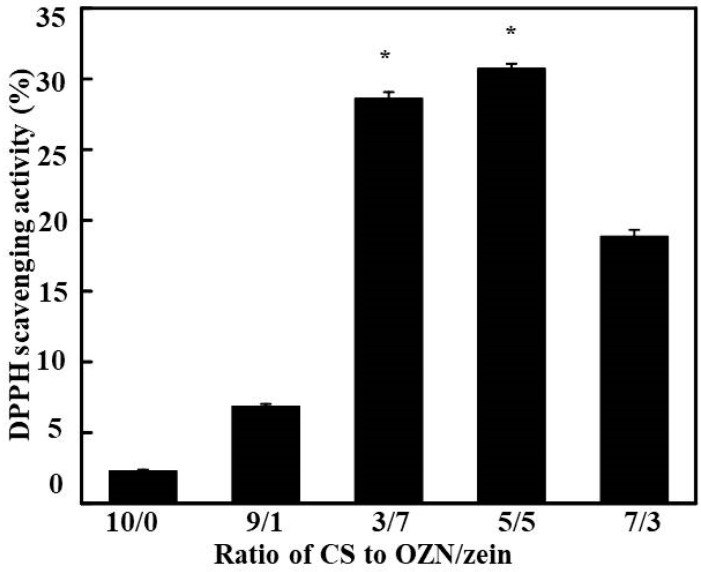
DPPH radical scavenging activity of different composite films on the 12th day (*: *p* < 0.05).

**Table 1 antioxidants-08-00391-t001:** Orthogonal test level of ultrasound-assisted extraction of orange-peel oil (OPO).

Level	Factor		
	Ultrasonic Power (W)	Ultrasonic Time (min)	Orange Peels/Ethyl Acetate Solution
1	80	30	1:6
2	90	40	1:8
3	100	50	1:10

**Table 2 antioxidants-08-00391-t002:** The result of an orthogonal test for OPO yield.

Serial Number	Ultrasonic Power (W)	Time (min)	Orange Peels/Ethyl Acetate Solution (g/mL)	Viscosity (MPa/s)	Yield of OPO (%)
1	80	30	1:6	5.79 ± 0.41 ^a^	1.76 ± 0.45 ^a^
2	80	30	1:8	9.23 ± 0.72 ^b^	2.32 ± 0.78 ^b^
3	80	30	1:10	8.44 ± 0.62 ^b^	1.88 ± 0.53 ^a^
4	80	40	1:6	6.32 ± 0.56 ^a^	1.85 ± 0.52 ^a^
5	80	40	1:8	10.18 ± 0.84 ^b^	2.56 ± 0.84 ^c^
6	80	40	1:10	9.06 ± 0.69 ^b^	2.02 ± 0.63 ^d^
7	80	50	1:6	6.33 ± 0.57 ^a^	1.80 ± 0.50 ^a^
8	80	50	1:8	9.88 ± 0.82 ^b^	2.41 ± 0.74 ^b^
9	80	50	1:10	8.59 ± 0.64 ^b^	1.98 ± 0.60 ^a^
10	90	30	1:6	6.12 ± 0.52 ^a^	1.82 ± 0.51 ^a^
11	90	30	1:8	10.09 ± 0.82 ^b^	2.44 ± 0.75 ^b^
12	90	30	1:10	8.47 ± 0.63 ^b^	1.95 ± 0.59 ^a^
13	90	40	1:6	6.21 ± 0.53 ^a^	1.90 ± 0.56 ^a^
14	90	40	1:8	10.92 ± 0.88 ^b^	2.78 ± 0.86 ^d^
15	90	40	1:10	9.12 ± 0.71 ^b^	2.32 ± 0.70 ^b^
16	90	50	1:6	6.18 ± 0.53 ^a^	1.85 ± 0.82 ^a^
17	90	50	1:8	10.49 ± 0.82 ^b^	2.63 ± 0.80 ^d^
18	90	50	1:10	9.68 ± 0.79 ^b^	2.26 ± 0.89 ^b^
19	100	30	1:6	5.99 ± 0.49 ^a^	1.79 ± 0.50 ^a^
20	100	30	1:8	9.62 ± 0.77 ^b^	2.32 ± 0.70 ^b^
21	100	30	1:10	8.52 ± 0.65 ^b^	1.92 ± 0.58 ^a^
22	100	40	1:6	6.16 ± 0.52 ^a^	1.85 ± 0.52 ^a^
23	100	40	1:8	9.79 ± 0.80 ^b^	2.55 ± 0.76 ^b^
24	100	40	1:10	9.35 ± 0.78 ^b^	2.14 ± 0.67 ^b^
25	100	50	1:6	6.04 ± 0.51 ^a^	1.81 ± 0.50 ^a^
26	100	50	1:8	9.68 ± 0.79 ^b^	2.43 ± 0.75 ^b^
27	100	50	1:10	8.76 ± 0.68 ^b^	2.09 ± 0.65 ^d^

Notes: ^a,b,c,d^ Means with different letters within a column indicate significant differences (*p* ≤ 0.05).

**Table 3 antioxidants-08-00391-t003:** Light transmission and color properties different CS/OZN composite films.

CS:OZN (*w/w*)	Light Transmission (%)	L*	a*	b*	ΔE
10:0	93.07 ± 1.32 ^b^	92.42 ± 1.12 ^a^	−1.06 ± 0.27 ^a^	1.74 ± 0.28 ^a^	------
9:1	89.54 ± 1.22 ^a^	90.21 ± 1.01 ^b^	−0.20 ± 0.12 ^c^	1.07 ± 0.27 ^c^	1.44 ± 0.23 ^a^
7:3	88.42 ± 1.19 ^a^	89.19 ± 0.96 ^a^	−0.17 ± 0.11 ^b^	1.29 ± 0.33 ^b^	1.36 ± 0.19 ^b^
5:5	87.29 ± 1.12 ^b^	88.02 ± 0.89 ^c^	−1.19 ± 0.32 ^b^	2.10 ± 0.43 ^a^	2.02 ± 0.46 ^c^
3:7	86.37 ± 1.08 ^a^	86.19 ± 0.77 ^a^	−0.79 ± 0.22 ^a^	2.51 ± 0.52 ^a^	2.35 ± 0.53 ^a^

Notes: CS:OZN (*w/w*) is the mass ratio of CS and OZN, light transmission was performed in the visible bright region, L*, a*, b*, ΔE represent lightness, red–green, yellow–blue, total color difference, respectively. Means with different letters within a column indicate significant differences (*p* ≤ 0.05).

**Table 4 antioxidants-08-00391-t004:** Properties of different CS/OZN composite films.

CS:OZN (*w/w*)	Thickness (mm)	TS (MPa)	EAB (%)	OP × 10^−14^ (cm^3^/m s Pa)	WVP (×10^−11^g/m s Pa)
10:0	0.122 ± 0.024 ^a^	9.42 ± 1.06 ^a^	25.50 ± 2.15 ^a^	2.78 ± 0.81 ^a^	6.62 ± 1.06 ^a^
9:1	0.154 ± 0.052 ^b^	10.21 ± 1.34 ^b^	26.48 ± 2.36 ^a^	2.30 ± 0.72 ^b^	4.45 ± 0.90 ^b^
7:3	0.142 ± 0.044 ^b^	12.19 ± 1.98 ^c^	29.14 ± 2.49 ^b^	1.91 ± 0.69 ^c^	3.78 ± 0.82 ^c^
5:5	0.137 ± 0.036 ^a^	12.19 ± 1.97 ^c^	30.91 ± 2.52 ^d^	1.80 ± 0.61 ^c^	3.02 ± 0.74 ^c^
3:7	0.129 ± 0.026 ^a^	10.12 ± 1.32 ^b^	35.65 ± 2.99 ^e^	1.79 ± 0.60 ^c^	2.15 ± 0.58 ^d^

Notes: Thickness of each composite film was measured by a thickness gauge, TS; EAB represents tensile strength and elongation at break of composite films; OP, WVP represent Oxygen and Water vapor transmittance. Means with different letters within a column indicate significant differences (*p* ≤ 0.05).

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
