# Peer review of "Preparation and Characterization of Corn Starch Bio-Active Edible Packaging Films Based on Zein Incorporated with Orange-Peel Oil"

_antioxidants, 2019, doi:10.3390/antiox8090391_

Round 1
Reviewer 1 Report
The abstract should be a summary of the main findings rather than a description of the methods which is currently around half of the content. There is also some repetition which should be avoided. There are eight mentions of D-limonene, please find a suitable abbreviation for the components so this is not overly repeated. For example, CS for corn starch, CZ for corn zein, LZN for limonene-zein nanocapsules etc.
Since the D-isomer of limonene is the main type occurring in nature, it would be acceptable to just refer to this as limonene.
Line 39: this is mainly due to the hydrophilic nature of starch.
Line 41: who is “They”?
Line 52: what “synthetic polymers” and what “toxic substances”? This needs better context or delete this sentence.
In general, the introduction should be revised for better context of the work presented in the manuscript. It reads as quite disjointed and there is limited discussion of the use and properties of limonene. There are many articles on this substance and the other citrus oil sources.
Methods
Line 101: “Content of D-limonene in essential oil was evaluated by GC analysis.” What were the conditions? This needs a better description. The extraction method described would also extract out the other citrus oils, not just the limonene. There are also no results presented regarding the GC analysis and the composition of the extracted oil.
What is the purpose of the turbidity test (2.4)? In the context of the manuscript, there does not seem to be justification for this test.
Line 129: why did you use 252 nm?
Line 139: why did you choose 55 degrees for drying? Citrus oils are volatile and they may have evaporated.
The schematic in Figure 1 is incomplete, you need to add the step for film formation.
Line 182: Did you use ATR for surface scanning? Please provide the details.
Results
Section 3.1 describes the extraction based on the yield but there is no evidence to show the quality of the oil. I doubt that it is pure limonene but rather orange-peel oil. The fact that you have a colored oil is evidence that there are other compounds in the extract since pure limonene is colorless.
Lines 288-291 are not discussion but introduction/theory. This should be revised in context with the results. Similarly lines 314-316, 356, 372, 411, etc.
Line 293: what is the evidence for the unsaturated double bond formation? This is not even mentioned in the FTIR results.
Line 320: you also added glycerine as a plasticizer (line 137), how do nanocapsules have a plasticizing effect? Is there any free limonene, or surface limonene that could add this effect? Are the capsules intact? I note the discussion in section 3.8 would suggest “explosive release”.
Table 4: please add the units for OP and WVP.
SEM figure 5: the resolution is too low to suggest uniform distribution of the nanocapsules.
Line 419: D-limonene contains flavonoids? You mention correctly that D-limonene is a monoterpene in line 415. I think you mean that orange-peel oil contains flavonoids. Please check and revise.
One key thing missing from this work is the influence of moisture/water on the integrity of the films. This will have a major impact on the target food for packaging of this material and should be either tested or mentioned that these films may only be suitable for dry foods. The conclusions line 444-446 are very general and the data does not support this statement.
Overall I have concerns with the term D-limonene. There is no evidence shown that you have extracted this compound from the oil, even though it may be a major component. You mention GC analysis but this is never shown. Please consider changing this to orange-peel oil throughout, or otherwise prove that limonene is extracted.
Author Response
Comments and Suggestions for Authors 1:
The abstract should be a summary of the main findings rather than a description of the methods which is currently around half of the content. There is also some repetition which should be avoided. There are eight mentions of D-limonene, please find a suitable abbreviation for the components so this is not overly repeated. For example, CS for corn starch, CZ for corn zein, LZN for limonene-zein nanocapsules etc.
Re: Yes, you are right, abstract was revised. I changed D-limonene into orange-peel oil in our paper. So we used CS stand for corn starch, OZN stand for orange-peel oil -zein nanocapsules. While zein is easy to write, do not need to change it. Thanks.
Since the D-isomer of limonene is the main type occurring in nature, it would be acceptable to just refer to this as limonene.
Re: I changed D-limonene into orange-peel oil in the manuscript. Thanks.
Line 39: this is mainly due to the hydrophilic nature of starch.
Re: Corrected. Thanks.
Line 41: who is “They”?
Re: Corrected. I pointed it clearly. Thanks.
Line 52: what “synthetic polymers” and what “toxic substances”? This needs better context or delete this sentence.
Re: Revised. Thanks.
In general, the introduction should be revised for better context of the work presented in the manuscript. It reads as quite disjointed and there is limited discussion of the use and properties of limonene. There are many articles on this substance and the other citrus oil sources.
Re: Yes, we added some contents in this section. Thanks.
Methods
Line 101: “Content of D-limonene in essential oil was evaluated by GC analysis.” What were the conditions? This needs a better description. The extraction method described would also extract out the other citrus oils, not just the limonene. There are also no results presented regarding the GC analysis and the composition of the extracted oil.
Re: Because the extraction method will extract some other citrus oils, and limonene is the main constituent of orange-peel oil(Wilkins, M.R.; Suryawati, L.; Maness, N.O.; Chrz, D. Ethanol production by Saccharomyces cerevisiae and Kluyveromyces marxianusin the presence of orange-peel oil. World J. Microbiol. Biotechnol. 2007, 23, 1161-1168.)(Ozturk, B.; Winterburn, J.; Gonzalez-Miquel, M. Orange peel waste valorization through limonene extraction using bio-based solvents. Biochem. Eng. J. 2019, 151, 107298.)( Esteban, J.; Ladero, M. Food waste as a source of value‐added chemicals and materials: abiorefinery perspective. Int.J.FoodSci.Technol. 2018, 53, 1–14), so I changed D-limonene into orange-peel oil in the manuscript.
The yield of orange-peel oil was evaluated by the formula which was widely used (Ozturk B.; Winterburn J.; Gonzalez-Mique M. Orange peel waste valorisation through limonene extraction using bio-based solvents, Biochem. Eng. J. 2019, 151, 107298).
( Orange-peel oil (OPO) yield (%)=×100 ) .Thanks.
What is the purpose of the turbidity test (2.4)? In the context of the manuscript, there does not seem to be justification for this test.
Re: In order to predict stability of orange-peel oil-zein (OZN) nanocapsules, we used turbidity test (Li P.H; Lu W.C. Effects of storage conditions on the physical stability of d-limonene nanoemulsion. Food Hydrocolloids, 2016, 53, 218-224.)( González-Reza, R.M.; Quintanar-Guerrero, D.; Real-López, A.D. Effect of sucrose concentration and pH onto the physical stability of β-carotene nanocapsule. LWT, 2018, 90:354-361.). The increased turbidity may caused by flocculation, coalescence, and aggregation when the ratio was increased (Mirhosseini, H.; Tan, C.P.; Hamid, N.S.A.; Yusof, S. Optimization of the contents of Arabic gum, xanthan gum and orange oil affecting turbidity, average particle size, polydispersity index and density in orange beverage emulsion. Food Hydrocolloid. 2008, 22, 1212-1223.). Therefore, it can further prevent flocculation by mix the corn starch fully. Thanks.
Line 129: why did you use 252 nm?
Re: The main component of orange-peel oil is limonene, and the maximum absorption wavelength of limonene is 252 nm, according to the related literature (Evageliou V , Saliari D . Limonene encapsulation in freeze dried gellan systems. Food Chemistry, 2017, 223(Complete):72-75.)( Li, P.H.; Lu, W.C. Effects of storage conditions on the physical stability of d-limonene nanoemulsion. Food Hydrocolloids, 2016, 53, 218-224.). Thanks.
Line 139: why did you choose 55 degrees for drying? Citrus oils are volatile and they may have evaporated.
Re: Sorry, we revised the wrong description. The high temperature may accelerate the release of orange-peel oil. Therefore, we used 35 degrees (Evageliou V., Saliari .. Limonene encapsulation in freeze dried gellan systems. Food Chemistry, 2017, 223:72-75.)( Li P H , Lu W C . Effects of storage conditions on the physical stability of d-limonene nanoemulsion. Food Hydrocolloids, 2016, 53, 218-224.). Although the temperature is not too low, the experimental results are better. Because zein has a good encapsulation effect, which can improve the stability of volatile essential oils (Hao, L.; Dongfeng, W.; Chengzhen. L.; Junxiang Z.;Minghao F.; Xun, S.;Teng, W.; Ying, X.; Yanping, C. Fabrication of stable zein nanoparticles coated with soluble soybean polysaccharide for encapsulation of quercetin. Food Hydrocolloids. 2019, 87, 342-351.)( Su, J.; Guo, Q.; Mao, L.; Gao, Y.; Yuan, F. Effect of gum arabic on the storage stability and antibacterial ability of β-lactoglobulin stabilized D-D-limonene emulsion. Food Hydrocolloid. 2018, 84, 75-83.). Our aim is to develop a novel zein-based core-shell nanoencapsules for protection and control release of limonene.( Chen, Y.; Shu, M.; Yao, X.; Wu, K.; Zhang, K.; He, Y.; Nishinari, K.; Phillips, G.O.; Yao, X.; Jiang, F. Effect of zein-based microencapsules on the release and oxidation of loaded limonene. Food Hydrocolloid. 2018, 84, 330-336. ). Thanks.
The schematic in Figure 1 is incomplete, you need to add the step for film formation.
Re: Added. Thanks.
Line 182: Did you use ATR for surface scanning? Please provide the details.
Re: Sorry, we only used AFM to test the surface. Thanks.
Results
Section 3.1 describes the extraction based on the yield but there is no evidence to show the quality of the oil. I doubt that it is pure limonene but rather orange-peel oil. The fact that you have a colored oil is evidence that there are other compounds in the extract since pure limonene is colorless.
Re: Yes, you are right, we changed the wrong description. Thanks.
Lines 288-291 are not discussion but introduction/theory. This should be revised in context with the results. Similarly lines 314-316, 356, 372, 411, etc.
Re: Yes, some parts were deleted. Thanks.
Line 293: what is the evidence for the unsaturated double bond formation? This is not even mentioned in the FTIR results.
Re: I rewrote this part. Thanks.
Line 320: you also added glycerine as a plasticizer (line 137), how do nanocapsules have a plasticizing effect? Is there any free limonene, or surface limonene that could add this effect? Are the capsules intact? I note the discussion in section 3.8 would suggest “explosive release”.
Re: In order to increase in the flexibility of the composite film, we added glycerin as the plasticizer after mixing OZN and corn starch together.(Paolicelli, P.; Petralito, S.; Varani, G. Effect of glycerol on the physical and mechanical properties of thin gellan gum films for oral drug delivery. Int. J. Pharm. 2018, 547, 226-234.). Thanks.
Table 4: please add the units for OP and WVP.
Re: Added. Thanks.
SEM figure 5: the resolution is too low to suggest uniform distribution of the nanocapsules.
Re: Figure 5 was replaced. Thanks.
Line 419: D-limonene contains flavonoids? You mention correctly that D-limonene is a monoterpene in line 415. I think you mean that orange-peel oil contains flavonoids. Please check and revise.
Re: Corrected. Thanks.
One key thing missing from this work is the influence of moisture/water on the integrity of the films. This will have a major impact on the target food for packaging of this material and should be either tested or mentioned that these films may only be suitable for dry foods. The conclusions line 444-446 are very general and the data does not support this statement.
Re: Yes, the conclusion was changed to be mainly suitable for dry matter due to the hydrophobicity of zein (Chen, H;. Zhong, Q. A novel method of preparing stable zein nanoparticle dispersions for encapsulation of peppermint oil, Food Hydrocolloid. 2015, 43,593–602.), and further experiments are needed. Thanks.
Overall I have concerns with the term D-limonene. There is no evidence shown that you have extracted this compound from the oil, even though it may be a major component. You mention GC analysis but this is never shown. Please consider changing this to orange-peel oil throughout, or otherwise prove that limonene is extracted.
Re: I changed D-limonene into orange-peel oil. Thanks.
Reviewer 2 Report
The paper reports the preparation of films made up of starch and zein nanocapsules loaded with limonene.
The paper is well-structured, and the reader has no problem to follow the work step by step. My feeling is that these materials have a potential in terms of application. The encapsulation of limonene as an antioxidant is a topic which fits with the journal.
My main concern is that it was not obvious to determine which part of the work was brand new and original by reading the introduction of the paper. It is mentioned that zein nanoparticules were already prepared. Nevertheless, when I read the paper for the first time, I got the feeling that the preparation of zein nanocapsules loaded with limonene was original because no citation was given. Obviously, this feeling was wrong because these loaded nanocapsules were already reported as shown by reference 60, which was cited later on in the paper ( Effect of zein-based microencapsules on the release and oxidation of loaded limonene, Food Hydrocolloids 2018, vol 84, pages 330–336, DOI: 10.1016/j.foodhyd.2018.05.049). It is essential to clearly cite previous works and to compare the differences, if any, with the new work. The last part of the paper, the preparation of the films made up of starch and the zein-loaded nanocapsules is original, at least to the best of my knowledge.
The text must be revised to take this comment into account. I consider this point as major but the authors should be able to revise their manuscript.
As minor remarks, the text is not always very clear for me.
For instance, in line 66, it is noted: “to encapsulate D-limonene with emulsions systems is an effective method to improve the physical stability”. I don’t follow what mean the authors by physical stability. In the previous sentence, they explained that the goal is to overcome oxidative degradation, which is an improvement of chemical stability.
In line 314, 314, it is noted: “Mechanical properties are usually characterized by elongation at break and tensile strength, reflecting the plasticity and power of the material, respectively”. What is the power of the material? I recommend using the common vocabulary used in the state of the art in the fields of materials.
In line 372, it is noted: “Infrared spectroscopy is a tool that can be used to study chemical bonds and compatibility” What is the connection between compatibility and Infrared spectroscopy.
In line 32, the authors mention the non-pollution to the environment by biodegradable materials. The authors are maybe optimistic here and this claim can not be true for all biodegradable materials. Indeed, in the case of slow degradation, the environment can be contaminated with partially degraded or fragmented materials, which could have a negative impact even if this is a temporary impact. Of course, this discussion is out of the scope of this paper. I just recommend avoiding questionable assertions.
Author Response
Comments and Suggestions for Authors 2:
The paper reports the preparation of films made up of starch and zein nanocapsules loaded with limonene. The paper is well-structured, and the reader has no problem to follow the work step by step. My feeling is that these materials have a potential in terms of application. The encapsulation of limonene as an antioxidant is a topic which fits with the journal.
My main concern is that it was not obvious to determine which part of the work was brand new and original by reading the introduction of the paper. It is mentioned that zein nanoparticules were already prepared. Nevertheless, when I read the paper for the first time, I got the feeling that the preparation of zein nanocapsules loaded with limonene was original because no citation was given. Obviously, this feeling was wrong because these loaded nanocapsules were already reported as shown by reference 60, which was cited later on in the paper ( Effect of zein-based microencapsules on the release and oxidation of loaded limonene, Food Hydrocolloids 2018, vol 84, pages 330–336, DOI: 10.1016/j.foodhyd.2018.05.049). It is essential to clearly cite previous works and to compare the differences, if any, with the new work. The last part of the paper, the preparation of the films made up of starch and the zein-loaded nanocapsules is original, at least to the best of my knowledge.
Re: I added some contents in the introduction section. Although the preparation of zein nanocapsules containing limonene is not original, but little research has done it. Due to its volatility, limonene produced a composite membrane that lost most of the limonene during the baking phase (Ren, S.; Trevino, E.; Dubé, M.A. CopolymerizationofLimonenewithn-Butyl Acrylate,Macromol.React.Eng. 2015, 9, 339–349.). However, some people use their volatility to form a composite film by plasma polymerization. (Gerchman, D.; Bones, B.; Pereira, M.B. Thin film deposition by plasma polymerization using d-limonene as a renewable precursor. Prog. Org. Coat. 2019, 129, 133-139.). In addition, this paper innovatively combines corn starch with zein nanocapsules loaded with limonene. The good film-forming properties of corn starch and the sustained-release properties of nanocapsules work synergistically to prolong the time of action of limonene.
The text must be revised to take this comment into account. I consider this point as major but the authors should be able to revise their manuscript.
Re: Corrected. Thanks.
As minor remarks, the text is not always very clear for me. For instance, in line 66, it is noted: “to encapsulate D-limonene with emulsions systems is an effective method to improve the physical stability”. I don’t follow what mean the authors by physical stability. In the previous sentence, they explained that the goal is to overcome oxidative degradation, which is an improvement of chemical stability.
Re: Limonene as a main content of OPO has strong volatility and is easily oxidized after being volatilized into air. The encapsulation can achieve a sustained release effect, thereby reducing its volatilization rate ( Hirota, R.; Roger, N.N.; Nakamura, H. Anti-inflammatory effects of limonene from yuzu (Citrus junos Tanaka) essential oil on eosinophils. Journal of Food Science, 2010, 3, 75.)( Doran, A.L.; Morden, W.E.; Dunn, K.; Edwards-Jones, V. Vapour–phase activities of essential oils against antibiotic sensitive and resistant bacteria including MRS. Lett. Appl. Microbiol., 2010, 48, 387-392.). Thanks.
In line 314, 314, it is noted: “Mechanical properties are usually characterized by elongation at break and tensile strength, reflecting the plasticity and power of the material, respectively”. What is the power of the material? I recommend using the common vocabulary used in the state of the art in the fields of materials.
Re: I used “strength of materials”. Thanks.
In line 372, it is noted: “Infrared spectroscopy is a tool that can be used to study chemical bonds and compatibility” What is the connection between compatibility and Infrared spectroscopy.
Re: The compatibility mentioned here refers to the interaction between substances. Infrared spectroscopy (IR) has an exclusively great significance in investigations of interactions among materials. The occurring interactions are discovered and proved by IR spectroscopy with the following important characteristics: appearance of new IR absorption band(s), broadening of band(s), alteration in intensity ( G.N. Kalinkova, Inter. J. Pharm. 1999, 187, 1–15.). The potential physical and chemical interactions among materials can affect the chemical nature, the stability. ( Rojek, B.; Wesolowski, M.; Suchacz, B. Detection of compatibility between baclofen and excipients with aid of infrared spectroscopy and chemometry. Spectroc. Acta Pt. A-Molec. Biomolec. Spectr. 2013, 532-538.) (Rojek, B., Wesolowski, M. Fourier transform infrared spectroscopy supported by multivariate statistics in compatibility study of atenolol with excipients. Vib. Spectrosc. 2016, 86, 190-197.). Thanks.
In line 32, the authors mention the non-pollution to the environment by biodegradable materials. The authors are maybe optimistic here and this claim can not be true for all biodegradable materials. Indeed, in the case of slow degradation, the environment can be contaminated with partially degraded or fragmented materials, which could have a negative impact even if this is a temporary impact. Of course, this discussion is out of the scope of this paper. I just recommend avoiding questionable assertions.
Re: Corrected. Thanks.
Round 2
Reviewer 1 Report
The authors have thoroughly and adequately addressed my comments and suggestions.
Reviewer 2 Report
The authors took into account my remarks and the i recommend the publication of the revised version of the manuscript.